# Efficacy of Thermal Ablation for Small-Size (0–3 cm) versus Intermediate-Size (3–5 cm) Colorectal Liver Metastases: Results from the Amsterdam Colorectal Liver Met Registry (AmCORE)

**DOI:** 10.3390/cancers15174346

**Published:** 2023-08-31

**Authors:** Madelon Dijkstra, Susan van der Lei, Robbert S. Puijk, Hannah H. Schulz, Danielle J. W. Vos, Florentine E. F. Timmer, Hester J. Scheffer, Tineke E. Buffart, M. Petrousjka van den Tol, Birgit I. Lissenberg-Witte, Rutger-Jan Swijnenburg, Kathelijn S. Versteeg, Martijn R. Meijerink

**Affiliations:** 1Department of Radiology and Nuclear Medicine, Amsterdam University Medical Centers, Location VUmc, 1081 HV Amsterdam, The Netherlands; m.dijkstra3@amsterdamumc.nl (M.D.); mr.meijerink@amsterdamumc.nl (M.R.M.); 2Department of Radiology and Nuclear Medicine, Noordwest Ziekenhuisgroep, 1815 JD Alkmaar, The Netherlands; 3Department of Medical Oncology, Amsterdam University Medical Centers, Location VUmc, Cancer Center Amsterdam, 1081 HV Amsterdam, The Netherlands; 4Department of Surgery, Medical Center Leeuwarden, 8934 AD Leeuwarden, The Netherlands; 5Department of Epidemiology and Data Science, Amsterdam University Medical Centers, Location VUmc, Vrije Universiteit Amsterdam, 1081 HV Amsterdam, The Netherlands; 6Department of Surgery, Amsterdam University Medical Centers, Location VUmc, Cancer Center Amsterdam, 1081 HV Amsterdam, The Netherlands

**Keywords:** colorectal liver metastases (CRLM), microwave ablation (MWA), radiofrequency ablation (RFA), intermediate-size

## Abstract

**Simple Summary:**

Thermal ablation is widely recognized as the standard of care for small-size (≤3 cm) colorectal liver metastases (CRLM) that are difficult to resect. The purpose of this comparative series was to analyze outcomes for intermediate-size (3.1–5 cm) versus small-size CRLM. In total, 280 patients undergoing 347 procedures between December 2000 and November 2021 were included. No significant difference between patients with small- versus intermediate-size CRLM was found in a comparison of overall survival. Per-tumor analysis showed that local control (LC) was superior in the small-size group. Nevertheless, the 1-, 3-, and 5-year LC for intermediate-size CRLM was still 93.9%, 85.4%, and 81.5%, and technical efficacy improved over time. In conclusion, thermal ablation for intermediate-size unresectable CRLM is safe and induces long-term local control in the vast majority of tumors.

**Abstract:**

Purpose: Thermal ablation is widely recognized as the standard of care for small-size unresectable colorectal liver metastases (CRLM). For larger CRLM safety, local control and overall efficacy are not well established and insufficiently validated. The purpose of this comparative series was to analyze outcomes for intermediate-size versus small-size CRLM. Material and methods: Patients treated with thermal ablation between December 2000 and November 2021 for small-size and intermediate-size CRLM were included. The primary endpoints were complication rate and local control (LC). Secondary endpoints included local tumor progression-free survival (LTPFS) and overall survival (OS). Results: In total, 59 patients were included in the intermediate-size (3–5 cm) group and 221 in the small-size (0–3 cm) group. Complications were not significantly different between the two groups (*p* = 0.546). No significant difference between the groups was found in an overall comparison of OS (HR 1.339; 95% CI 0.824–2.176; *p* = 0.239). LTPFS (HR 3.388; *p* < 0.001) and LC (HR 3.744; *p* = 0.004) were superior in the small-size group. Nevertheless, the 1-, 3-, and 5-year LC for intermediate-size CRLM was still 93.9%, 85.4%, and 81.5%, and technical efficacy improved over time. Conclusions: Thermal ablation for intermediate-size unresectable CRLM is safe and induces long-term LC in the vast majority. The results of the COLLISION-XL trial (unresectable colorectal liver metastases: stereotactic body radiotherapy versus microwave ablation—a phase II randomized controlled trial for CRLM 3–5 cm) are required to provide further clarification of the role of local ablative methods for intermediate-size unresectable CRLM.

## 1. Introduction

Colorectal cancer (CRC) is the third most common form of cancer in the world. CRC has an incidence of almost 1.9 million patients per year and was responsible for 9.4% of all cancer related mortality in 2020 [1]. A leading cause of death in CRC patient is related to the development of colorectal liver metastases (CRLM), which occur in roughly 50% of all CRC patients [2,3,4,5]. The presence of CRLM is a lethal condition when untreated, with 5-year overall survival (OS) rates of 0–3% [6,7,8]. Systemic therapy alone improves 5-year OS to approximately 11% [6,7,8,9]. With 5-year OS rates of 40–55% for upfront resectable disease and approximately 33% for patients downstaged with systemic therapy, partial hepatectomy remains the current standard of care to treat superficially located and resectable CRLM; however, only 20–30% of patients are considered eligible [3,4,5,10,11,12,13,14,15].

Several radical intent thermal and non-thermal ablative therapies have gradually gained acceptance in the international guidelines to treat unresectable CRLM [16,17,18,19,20,21,22,23]. Unresectable disease is herein defined as inability to obtain R0 margins, inability to spare sufficient future liver remnant volume and function, reduced general health status and/or major cardiopulmonary comorbidities, or presumed extensive adhesions caused by previous abdominal surgery [24]. The most utilized and researched thermal ablative energies are radiofrequency ablation (RFA) and microwave ablation (MWA), whereas evidence for non-thermal ablative methods such as irreversible electroporation (IRE) and stereotactic ablative body radiotherapy (SABR) ispiling.

Increased tumor sizes (>3 cm) are associated with exponentially reduced technical efficacy and shorter local tumor progression (LTP)-free survival [25,26,27,28,29,30,31]. A recent systematic review and meta-analysis by van Nieuwenhuizen et al. compared safety and efficacy of thermal ablation, IRE, and SABR for intermediate-size CRLM (3–5 cm) [32]. Per-patient local control ranged 22–89% (in eight series) following thermal ablation, and the results improved over time. Nonetheless, thermal ablation for unresectable intermediate-size tumors is currently still outside most of the international guidelines.

The suboptimal local efficacy emphasizes the necessity to further validate thermal ablation for intermediate-size CRLM. The aim of this Amsterdam Colorectal Liver Met Registry (AmCORE)-based study was to analyze efficacy of thermal ablation for small-size (0–3 cm) versus intermediate-size (3–5 cm) CRLM.

## 2. Materials and Methods

This single-center study was conducted at the Amsterdam University Medical Centers, the Netherlands, a tertiary referral medical center for gastrointestinal and hepatobiliary cancer. The prospectively maintained AmCORE database was used for data extraction, and data reporting is in accordance with the ‘Strengthening the Reporting of Observational studies in Epidemiology’ (STROBE) guideline [33]. The affiliated Institutional Review Board granted permission for the AmCORE database (METc 2021.0121).

### 2.1. Patient Selection and Data Collection

Per-patient and per-tumor data of patients undergoing thermal ablation for small-size (0–3 cm) and intermediate-size (3–5 cm) CRLM were identified and collected from the database, and were analyzed conformal to the SIO-DATECAN consensus document [34]. 

Patients with at least 1 tumor > 3 cm and ≤5 cm were included in the intermediate-size (3–5 cm) group, regardless of the concomitant presence of additional small-size CRLM. Patients with merely ablations for smaller-size tumors were included in the small-size (0–3 cm) group. If additional information was needed, recollecting of data was executed by retrospectively searching the hospital’s electronic patient database. Patients receiving thermal ablation alone for ≤5 cm CRLM were included. Patients receiving concomitant surgical resection, SABR, or IRE, and patients in whom follow-up was too short or insufficient, were excluded. If patients received multiple ablation sessions, only the initial procedure and tumors ablated in that specific session were taken into account regarding per-patient survival outcomes. 

### 2.2. Thermal Ablation Procedure

All patients with CRLM potentially suitable for local treatment were discussed by a multidisciplinary tumor board, attended by (interventional) radiologists, hepatopancreaticobiliary and/or oncological surgeons, medical oncologists, radiation oncologists, nuclear medicine physicians, gastroenterologists, and pathologists. Imaging included contrast enhanced computed tomography (ceCT), contrast enhanced magnetic resonance imaging (ceMRI), and [18F]-fluoro-2-deoxy-D-glucose (^18^F-FDG) positron emission tomography (PET)—CT scans, and it was assessed using the RECIST criteria [35]. 

Two experienced (defined as having performed and/or supervised > 100 procedures) interventional radiologists performed and/or supervised the ablations, and the treatment protocols were in accordance with the instructions for use as provided by the manufacturer and the CIRSE quality improvement guidelines [36]. Conformal to the CIRSE standards of practice on thermal ablation of liver tumors, the intended minimum tumor free ablation margin was >1 cm and the minimum realized tumor-free ablation margin to claim technical success was 5 mm [37,38]. Ablation zone margins were calculated with confirmation software with rigid 3D image-registration (Syngo Fusion, Siemens, Erlangen, Germany) directly after the ablation. A CT-guided percutaneous approach was preferred; laparoscopic and open procedures were reserved for cases where critical structures, such as the intestines, could not be distanced using pneumo- or hydrodissection. The RF3000 generator with expandable LeVeen electrodes (RFA; Boston Scientific, Marlborough, MA, USA), the RITA system with compatible expandable electrodes (RFA; AngioDynamics BV, Amsterdam, The Netherlands), the Evident system (MWA; Medtronic-Covidien, Minneapolis, MN, USA), the Emprint system (MWA; Medtronic-Covidien, Minneapolis, MN, USA), or the Solero (MWA; AngioDynamics BV, Amsterdam, The Netherlands) generators with compatible antennas were used for nearly all thermal ablation procedures. 

In accordance with national guidelines, the use of (neo)adjuvant systemic therapy was not routine [39]. Induction systemic therapy for downsizing to reduce procedural risk and neoadjuvant systemic therapy for patients with potentially worse tumor biology (multiple intrahepatic recurrences < 6 months) were excepted. Potentially insufficient ablation margins were treated with overlapping ablations of residual tumor tissue.

### 2.3. Follow-Up

A ceCT scan was performed <6 weeks after thermal ablation when the risk for residual disease was considered high. As recommended by national guidelines, ^18^F-FDG-PET CT scans were performed every 3–4 months in the first year, every 6 months in the second and third year, and every 12 months in the fourth and fifth year following thermal ablation [39]. LTP was described as a solid and unequivocally enlarging mass or as focal ^18^F-FDG PET avidity at the surface of the ablated tumor. Additional ceMRI or image-guided biopsies were performed in case of uncertainty.

### 2.4. Statistical Analysis

Baseline characteristics concerning per-patient and per-tumor data were compared between the two groups: small-size versus intermediate-size. Categorical characteristics were described as percentages of patients and compared using the Pearson chi-square test, except for dichotomous characteristics, where the Fisher’s exact test was used. Continuous characteristics were described as mean with standard deviation (SD) or median with interquartile range (IQR), and compared using the independent t-test or the Mann–Whitney U test. Complications were presented using Common Terminology Criteria for Adverse Events (CTCAE) 5.0 and analyzed using the chi-square test. Length of hospital stay was analyzed using the Mann–Whitney U test. Complications and length of hospital stay were both assessed per procedure. 

Primary endpoint LC (per tumor, allowing re-treatments) and secondary endpoints LTPFS (per tumor) and OS (per patient, from first local treatment), all defined as time-to-event from thermal ablation, were analyzed using Kaplan–Meier curves with a log-rank test [34]. In addition, primary endpoint LC was reviewed using Cox proportional hazards regression models, accounting for potential confounders in multivariable analysis. Potential confounders were first identified in the analysis of characteristics (*p* < 0.100), subsequently in univariable analysis (*p* < 0.100), and with use of the backward selection procedure included in multivariable analysis. Variables were considered as potential confounders when *p* < 0.050 in the final model. Variables were considered actual confounders when the regression coefficient in the Cox regression model for LC changed by >10% in the corrected model. Hazard ratio (HR) and 95 per cent confidence interval (95% CI) were calculated.

Statistical analyses were conducted in agreement with a biostatistician (BILW), and SPSS^®^ Version 28.0 (IBM^®^, Armonk, New York, NY, USA) [40] and R version 4.0.3. (R Foundation, Vienna, Austria) were used to perform the analyses [41].

## 3. Results

A total of 338 patients receiving thermal ablation alone were identified from the prospective AmCORE database. Eventually, 280 patients undergoing 347 procedures with 856 CRLM between December of 2000 and November of 2021 were included for further analyses (Figure 1).

### 3.1. Patient- and Disease-Related Characteristics

Patients with at least one tumor > 3 cm and ≤5 cm were included in the intermediate-size (3–5 cm) group (N = 59). Patients with merely ablations for smaller-size tumors were included in the small-size (0–3 cm) group (N = 221). Patient- and disease-related characteristics are presented in Table 1. Most patients in this cohort were male (69.3%). The mean age of this cohort was 65.6 years (SD 11.1). Comorbidities differed significantly between the small-size group and intermediate-size group. Patients with small-size CRLM presented less frequently with comorbidities compared to patients with intermediate-size CRLM (11.9% vs. 27.6%; *p* = 0.012). Disease-related characteristics concerning primary tumor location, molecular profile, and extrahepatic disease were well-balanced among the two groups. More patients in the small-size group were diagnosed with synchronous disease compared to the intermediate-size group (59.2% vs. 42.6%; *p* = 0.032). Median follow-up time after thermal ablation was 24.2 months in both groups.

### 3.2. Procedure- and Tumor-Related Characteristics

Procedures where at least one intermediate-size tumor > 3 cm and ≤5 cm was treated were included in the intermediate-size group (N = 60); the other procedures were included in the small-size (≤3 cm) group (N = 287). A total of 783 tumors were included in the small-size group and 73 tumors in the intermediate-size group. Table 2 shows the procedure and tumor-related characteristics. The total number of tumors treated in the same procedure was significantly higher for small-size versus intermediate-size CRLM (*p* < 0.001). Thermal ablation techniques and modalities were well-balanced over the two groups. No significant difference in approach was found between groups. Most patients received general anesthesia. The vast majority of ablation zones of small-size tumors showed margins > 5 mm (94.2%), whereas, for intermediate-size tumors, only 58.5% reached margins > 5 mm (*p* = 0.020). Median tumor size in the small-size group was 13.0 mm (IQR 8.0–20.0), and median tumor size in the intermediate-size group was 36.0 mm (IQR 33.0–40.5).

### 3.3. Complications and Length of Hospital Stay

The number and severity of complications was not significantly different between the small-size and intermediate-size groups (Table 3; *p* = 0.546). The complication rate was 33/221 (14.9%) of patients with small-size CRLM and 9/59 (15.3%) of patients with intermediate-size CRLM. One patient in the intermediate-size group had a grade 4 complication following open thermal ablation: post-procedural ileus and aspiration pneumonia with staphylococcus aureus bacteremia requiring intensive care unit admission. The median length of hospital stay was 1 day (IQR 1.0–4.0) in the small-size group compared to 4 days (IQR 1.0–5.0) in the intermediate-size group (*p* = 0.002).

### 3.4. Overall Survival (OS)

All patients receiving thermal ablation alone as the first local treatment were included in analysis of OS (Figure 2): 154 patients in the small-size group and 42 patients in the intermediate-size group. Median OS was 50.3 months in the whole cohort, 53.0 months for patients with small-size CRLM, and 40.7 months for patients with intermediate-size CRLM. In total, 74 out of 196 patients (37.8%) died during follow-up, 49 out of 154 (31.8%) in the small-size group and 25 out of 42 (37.8%) in the intermediate-size group. No significant difference between patients with small- and intermediate-size CRLM was revealed in the overall comparison of OS (HR 1.339; 95% CI 0.824–2.176; *p* = 0.239). Altogether, the 1-, 3-, and 5-year OS rates were 91.7%, 65.6%, and 37.1%, respectively. In the small-size group, the 1-, 3-, and 5-year OS rates were 91.8%, 68.1%, and 39.5%, respectively. In the intermediate-size group, the 1-, 3-, and 5-year OS rates were 91.6%, 59.0%, and 31.4%, respectively. Though a higher number of CRLM were present in the small-size group, univariable analysis did not identify the number of CRLM as potential confounder regarding OS (*p* = 0.84).

### 3.5. Local Tumor Progression-Free Survival (LTPFS) and Local Tumor Control (LC)

During follow-up, LTP developed in 91 of 856 tumors (10.6%); 71/783 (9.1%) were small-size tumors, and 20/73 (27.4%) were intermediate-size tumors (Figure 3A). LTPFS was superior in the small-size group compared to the intermediate-size group (HR 3.388; 95% CI 2.060–5.570; *p* < 0.001). In the small-size group, the 1-, 3-, and 5-year LPTFS rates were 92.5%, 88.1%, and 88.1%, respectively. In the intermediate-size group, the 1-, 3-, and 5-year LTPFS rates were 74.7%, 66.0%, and 66.0%, respectively. The results of LTPFS significantly improved over time. Comparing results of LC before 2010 and after 2010 for intermediate-size CRLM, a significant difference was found (HR 0.315; 95% CI 0.127–0.781; *p* = 0.013) in favor of tumors treated after 2010.

Eventual loss of LC at follow-up was reported in 24 out of 856 tumors (2.8%), 16 out of 783 (2.0%) small-size tumors, and 8 out of 73 intermediate-size tumors (11.0%) (Figure 3B). The 1-, 3-, and 5-year LC rates were 98.6%, 96.7%, and 94.0%, respectively, in the whole cohort, 99.1%, 97.8%, and 95.3% in the small-size group, and 93.9%, 85.4%, and 81.5% in the intermediate-size group. Compared to small-size CRLM, LC was significantly lower in intermediate-size CRLM (HR 5.383; 95% CI 2.303–12.584; *p* < 0.001). 

ASA, comorbidities, time to first diagnosis of CRLM, preprocedural chemotherapy, number of metastases, and margin size differed significantly when comparing baseline characteristics between the two groups and were included in univariable analysis (Table 4). Univariable analysis identified four potential associations with LC: gender (*p* = 0.025), age (*p* = 0.040), number of tumors (*p* < 0.001), and margin size (*p* = 0.008). The variables were included in multivariable analysis to analyze potential confounders associated with the two groups influencing LC. Gender (*p* = 0.008) and number of tumors (*p* = 0.003) were significant confounders in the multivariable analysis. The corrected HR for LC was still significantly worse for intermediate-size CRLM (HR 3.744; 95% CI 1.537–9.125; *p* = 0.004).

## 4. Discussion

Thermal ablation has emerged as a safe and effective treatment option to eradicate small-size, unresectable CRLM (≤3 cm). For larger CRLM, safety, local control, and overall efficacy are not well established and insufficiently validated. In this AmCORE-based study, patients with intermediate-size CRLM demonstrated lower LTPFS and LC compared to patients with small-size CRLM. During follow-up, LTP developed in 27.4% of intermediate-size tumors, and 5-year LTPFS was 66.0%. Though these results seem to validate thermal ablation for intermediate-size unresectable CRLM, the outcomes require further improvement before partial hepatectomy can be truly challenged for larger lesions. However, including repeat treatments, the vast majority of thermally ablated intermediate-size tumors were ultimately eradicated with a LC of approximately 80%.

Larger tumor size did not significantly affect complications, though patients had an increased length of hospital stay (median of 4 days versus 1 day for small-size CRLM). The difference in length of hospital stay may be caused by the difference in treatment approach, where 46.7% of the small-size group vs. 54.8% of the intermediate-size group were treated with an open approach. However, safety was not at risk, as the total complication rate was 15.3% for intermediate-size tumors compared to 14.9% for small-size tumors, and complications per grade did not significantly differ between groups. Interestingly, no significant difference in OS was observed between the small-size and the intermediate-size group. This suggests that thermal ablation may yield similar survival outcomes for both small- and intermediate-size CRLM.

In the current literature, increased tumor sizes over 3 cm are associated with exponentially reduced technical efficacy, leading to increased LTP rates following thermal ablation [25,26,27,28,29,30,31,43,44,45]. In a recent study assessing primary tumor sidedness and mutational status, subgroup analyses of intermediate-size (3–5 cm) CRLM (12.4% of 2101 tumor) showed a reduced local tumor progression-free survival (LTPFS) associated with increasing tumor volume [22]. In addition, a recent systematic review and meta-analysis by Nieuwenhuizen et al. compared safety and efficacy of thermal ablation, IRE, and SABR for intermediate-size CRLM [32]. Following thermal ablation, LTP was reported in up to 62% of patients with intermediate-size CRLM [25,46,47,48]. Mao et al. and Nielsen et al. [25,47] described comparable results to our series, with LTP rates of 25% and 27%, respectively. Only Bale et al. found lower LTP rates of intermediate-size CRLM compared to small-size CRLM (11.1% vs. 17.7%) [46]. Complications were not specifically reported for patients with intermediate-size CRLM by Nieuwenhuizen et al. or Bale et al.; however, the present complication rates did not vary from the total complication rates of the whole cohort of these series [32,46].

Consensus concerning local thermal and non-thermal ablative therapies for unresectable (intermediate-size) CRLM has not been reached, as a result of a lack of studies directly comparing RFA to MWA, SABR, or IRE [24,32]. RFA and MWA are currently widely adopted treatment techniques for small-size unresectable CRLM, given the safety profile and LC, and are now challenging surgical resection for upfront resectable CRLM ≤ 3 cm to prove non-inferiority in the COLLISION trial [16,24,49,50]. SABR has been suggested by the radiation oncology community as an alternative for limited number of intermediate-size unresectable CRLM, as it is associated with an excellent safety profile and acceptable LC rates that are potentially less affected by increased size [51,52,53,54]. However, higher complication rates associated with thermal ablation of local tumor recurrences after SABR should be taken into account when choosing the right treatment sequence [51]. In addition, a recent study by van Nieuwenhuizen et al., with potential residual confounding, showed superior LTPFS and LC of thermal ablation compared to SABR [51]. The ongoing phase II/III randomized controlled COLLISION-XL trial (NCT04081168) for unresectable intermediate-size CRLM, comparing SABR to MWA, should provide definitive answers [55]. At last, IRE recently arose as a non-thermal ablative method inducing permanent disruption of the cell membrane with the use of high-voltage electric pulses. This technique could be especially useful for CRLM adjacent to vascular and biliary structures, and it is also potentially less influenced by tumor volume as it represents a multi-electrode tumor-bracketing technique [56,57,58].

Important prognosticators of LTP are the peri-ablational safety margins, predicting technical success (A0 ablations), LTPFS, and LC. In this study, a minimum of 5 mm margin, and, if possible, over 10 mm, surrounding the tumor is suggested to obtain these A0 ablation margins [37,59,60,61,62]. For larger-size tumors, the preferred size of the tumor-free ablation zone is not only a trade-off between efficacy and safety, but also requires taking into account tumor perfusion, tumor boundaries, interstitial space porosity during heating, and the applied heat dosage in order to spare healthy surrounding parenchyma [63,64]. Multiple technical developments have been proposed to improve tumor visibility with accurate needle tracking and positioning, such as real-time navigation, image fusion, and computed tomography hepatic arteriography (CTHA) guidance of percutaneous ablation [37,59,60,61,62,65,66,67]. As discussed by Puijk et al. and confirmed by this study for intermediate-size CRLM, efficacy (LTPFS) has significantly improved over time [43]. Forthcoming technical improvements should further contribute to prevent insufficient treatment and provide even longer LTPFS and LC. Adequate A0 ablation margins are challenged by enlarged tumor sizes [68]. To achieve the above-mentioned margin sizes in intermediate-size CRLM, multiple electrodes may be used to increase the size of the ablation zone. However, the treatment strategy for thermal ablation of larger-size CRLM is frequently found to be operator dependent [68].

The relatively high number of tumors endorsed adequately powered statistical analyses, therefore strengthening this study. However, the non-randomized study design is a substantial limitation, which potentially induced selection bias and confounding. Additional multivariable analysis was performed to account for potential confounders; nonetheless, exclusion of all residual confounding is not assured. The concomitant presence of small-size CRLM in many intermediate-size group patients and the significantly higher number of CRLM in the small-size group may pose confounders for survival due to the fact that prognosis is not only influenced by size, but also by the number of CRLM. However, since both size and number represent parameters to quantify volumetric disease burden, this confounder is at least partially nullified given the inverse correlation between size and volume.

All patients were discussed in a multidisciplinary tumor board; therefore, the choice of treatment was based on local expertise, which may induce selection bias. In addition, inclusion of patients treated over >20 years ago may have led to population or historical bias. Furthermore, improved thermal ablation techniques, as well as the use of confirmation software and CTHA, have led to increased technical efficacy over the study period [43]. The technique of thermal ablation of patients treated in this study do not represent all present, universal thermal ablation techniques. As it is likely that operator-experience is strongly correlated to outcome, results cannot be automatically extrapolated to centers with more limited experience.

## 5. Conclusions

To conclude, LTPFS and LC were inferior when comparing thermal ablation for intermediate-size versus small-size CRLM. Nonetheless, the low complication rate, comparable OS, and the relatively high rate of eventual LC (80%) seem to validate thermal ablation for unresectable intermediate-size CRLM in high-volume dedicated centers. Further research is warranted to explore strategies to optimize local control for intermediate-size CRLM following treatment with thermal and non-thermal ablation techniques. 

Results from randomized controlled trials such as the COLLISION-XL trial (NCT04081168) (unresectable colorectal liver metastases: stereotactic body radiotherapy versus microwave ablation—a phase II randomized controlled trial for CRLM 3–5 cm) are required in order to provide clarification on the preferred local ablative method for intermediate-size unresectable CRLM.

## Figures and Tables

**Figure 1 cancers-15-04346-f001:**
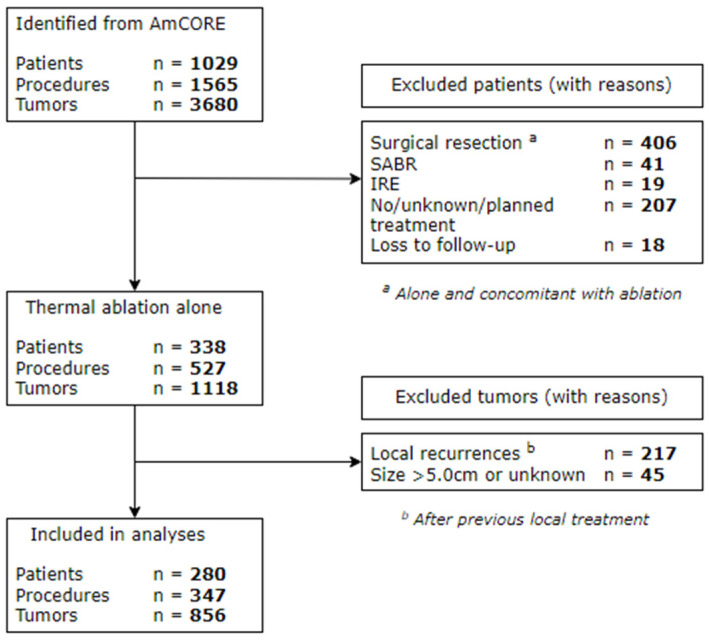
Flowchart of included and excluded patients.

**Figure 2 cancers-15-04346-f002:**
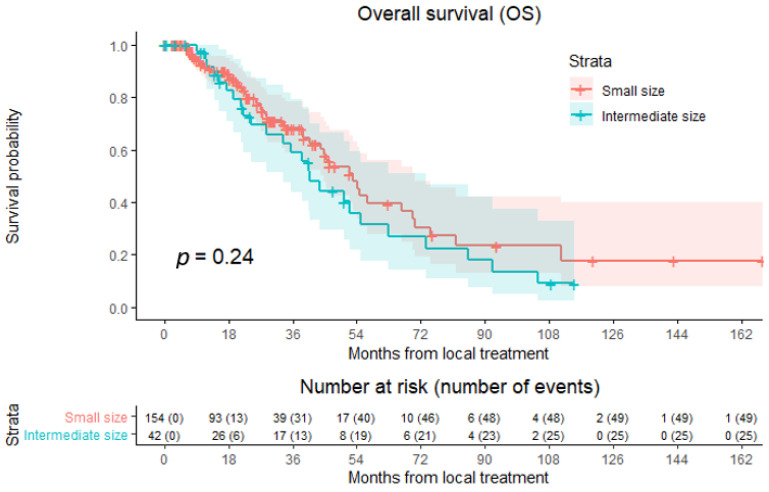
Kaplan–Meier curves of overall survival (OS) after thermal ablation of small-size CRLM (red) versus intermediate-size CRLM (green). Numbers at risk (number of events) are per patient. Overall comparison log-rank test, *p* = 0.240.

**Figure 3 cancers-15-04346-f003:**
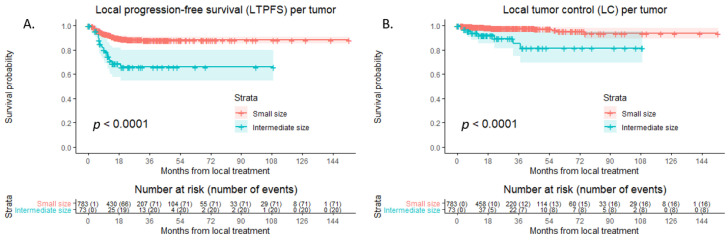
Kaplan–Meier curves of (**A**) local tumor progression-free survival (LTPFS) and (**B**) local tumor control (LC) per tumor after thermal ablation of small-size CRLM (red) versus intermediate-size CRLM (green). Numbers at risk (number of events) are per tumor. Overall comparison log-rank (Mantel–Cox) test, (**A**) *p* < 0.001 and (**B**) *p* < 0.001. Death without local tumor progression (LTP) or loss of LC is censored.

**Table 1 cancers-15-04346-t001:** Patient- and disease-related characteristics.

		Total N = 280	Small N = 221	Intermediate N = 59	*p*-Value
**Patient-Related Characteristics**					
Gender	Male	69.3	67.9	74.6	
	Female	30.7	32.1	25.4	0.346 ^a^
Age (years)	Mean (SD)	65.6 (11.1)	65.3 (11.2)	66.8 (10.6)	0.365 ^b^
ASA physical status	1	6.5	6.9	5.3	
	2	69.8	72.9	57.9	
	3	23.3	19.7	36.8	
	4	0.4	0.5	0.0	0.055 ^c^
Comorbidities	None	49.8	52.1	41.4	
	Minimal	35.0	36.1	31.0	
	Major	15.2	11.9	27.6	0.012 ^c^
**Disease-related characteristics**					
Primary tumor location	Right-sided colon	21.8	21.7	22.0	
	Left-sided colon	47.1	48.0	44.1	
	Rectum	31.1	30.3	33.9	0.842 ^c^
Molecular profile	RASwt/mut/unknown	11.4/7.1/81.5	11.3/7.7/81.0	11.9/33.9/54.2	0.196 ^c^
	BRAFwt/mut/unknown	16.8/1.1/82.1	17.2/1.4/81.4	15.3/1.7/83.0	0.236 ^c^
	MSS/MSI/unknown	29.6/0.4/73.0	30.3/0.5/69.2	27.1/0.0/72.9	0.624 ^c^
Time interval to diagnosis CRLM	Metachronous	44.2	40.8	57.4	
	Synchronous	55.8	59.2	42.6	0.032 ^c^
Extrahepatic disease at first diagnosis of CRLM	No	93.2	93.1	93.6	
	Yes	6.8	6.9	6.4	1.000 ^c^

Categorical variables are reported as % of patients, continuous variables are reported as mean (SD), ^a^ = Fisher’s Exact Test, ^b^ = Independent *t*-Test, ^c^ = Pearson Chi-Square, ASA = American Society of Anesthesiologists score.

**Table 2 cancers-15-04346-t002:** Procedure- and tumor-related characteristics.

		**Total** **N = 347**	**Small** **N = 287**	**Intermediate** **N = 60**	***p*-Value**
**Procedure-Related Characteristics**					
Preprocedural chemotherapy	No	67.8	67.2	72.1	
	Yes	32.2	32.8	27.9	0.603 ^a^
Procedure number in course of treatment	1st	57.3	56.1	65.9	
	2nd–5th	40.9	41.9	34.1	
	>5th	1.7	2.0	0.0	0.352 ^b^
Number of tumors	1	50.1	46.2	77.3	
	2–5	39.5	41.9	22.7	
	>5	10.4	11.9	0.0	<0.001 ^b^
Ablation technique	RFA	34.9	33.0	47.7	
	MWA	65.1	67.0	52.3	0.063 ^a^
Ablation modality	RFA				
	RF3000™, LeVeen™	29.2	27.1	43.2	
	Cool-tip™	3.5	4.1	0.0	
	Starburst^®^ (RITA^®^)	1.2	1.0	2.3	
	Others	0.6	0.3	2.3	
	MWA				
	Evident™	2.1	2.0	2.3	
	Emprint™	54.0	55.9	40.9	
	Solero™	0.3	0.0	0.3	
	Others	9.1	9.2	9.1	0.268 ^b^
Approach	Open	30.3	28.7	40.9	
	Percutaneous	69.7	71.3	59.1	0.115 ^a^
Image-guidance technique	Conventional *	48.4	47.2	56.8	
	CTHA	51.6	52.8	43.2	0.261 ^a^
Anesthesia	Midazolam sedation	8.7	9.0	6.8	
	Propofol sedation	38.6	39.5	31.8	
	General anesthesia	52.8	51.5	61.4	0.471 ^b^
		**Total** **N = 856**	**Small** **N = 783**	**Intermediate** **N = 73**	***p*-Value**
**Tumor-Related Characteristics**					
Size (mm)	Median (IQR)	15.0 (9.0–22.0)	13.0 (8.0–20.0)	36.0 (33.0–40.5)	<0.001 ^c^
Margin size (mm)	0–5	6.5	5.8	14.5	
	>5	93.5	94.2	58.5	0.020 ^a^

Values are reported as % of patients and continuous variables are reported as median (IQR), * = intraoperative ultrasound or CT fluoroscopy, ^a^ = Fisher’s Exact Test, ^b^ = Pearson Chi-Square, ^c^ = Mann–Whitney U Test, RFA = radiofrequency ablation, MWA = microwave ablation, CTHA = CT hepatic arteriography.

**Table 3 cancers-15-04346-t003:** Complications and length of hospital stay (CTCAE) [42].

	Total N = 280	Small N = 221	Intermediate N = 59	*p*-Value
Complications				
Grade 1	3.6	3.6	3.4	
Grade 2	6.1	6.8	3.4	
Grade 3	5.0	4.1	8.5	
Grade 4	0.4	0.5	0.0	
Grade 5	0.0	0.0	0.0	0.546 ^a^
Length of hospital stay	1.0 (1.0–4.8)	1.0 (1.0–4.0)	4.0 (1.0–5.0)	0.002 ^b^

Values are reported as % of patients and median days (IQR), ^a^ = Pearson Chi-Square, ^b^ = Mann–Whitney U Test.

**Table 4 cancers-15-04346-t004:** Uni- and multivariable Cox regression analysis to detect variables associated with local tumor control (LC).

		Univariable Analysis		Multivariable Analysis	
		HR (95% CI)	*p*-Value	HR (95% CI)	*p*-Value
Size	Small	Reference	<0.001	Reference	0.004
	Intermediate	5.383 (2.303–12.584)		3.744 (1.537–9.125)	
**Patient-related characteristics**					
Gender	Male	Reference	0.025	Reference	0.008
	Female	2.497 (1.120–5.569)		2.980 (1.326–6.695)	
Age		1.043 (1.002–1.086)	0.040	1.027 (0.980–1.077)	0.266
ASA physical status	1	Reference	0.444		
	2	NA			
	3	NA			
	4	NA			
Comorbidities	None	Reference	0.211		
	Minimal	0.651 (0.251–1.685)			
	Major	1.860 (0.641–5.394)			
**Disease-related characteristics**					
Primary tumor location	Right-sided colon	Reference	0.793		
	Left-sided colon	0.901 (0.339–2.392)			
	Rectum	0.673 (0.205–2.208)			
First diagnosis of CRLM	Metachronous	Reference	0.122		
	Synchronous	0.508 (0.215–1.199)			
Extrahepatic disease at first diagnosis of CRLM	No	Reference	0.345		
	Yes	2.035 (0.465–8.899)			
**Procedure-related characteristics**					
Preprocedural chemotherapy	No	Reference	0.287		
	Yes	0.633 (0.272–1.470)			
Procedure number in course of treatment	1st	Reference	0.223		
	2nd–5th	2.044 (0.911–4.586)			
	>5th	NA			
Number of tumors	1	Reference	<0.001	Reference	0.003
	2–5	0.247 (0.098–0.620)		0.281 (0.109–0.721)	
	>5	0.142 (0.046–0.437)		0.183 (0.057–0.588)	
Ablation technique	RFA	Reference	0.916		
	MWA	0.954 (0.403–2.263)			
Approach	Open	Reference	0.260		
	Percutaneous	1.264 (0.841–1.900)			
Image-guidance technique	Conventional *	Reference	0.832		
	CTHA	1.097 (0.465–2.585)			
Anesthesia	Midazolam sedation	Reference	0.116		
	Propofol sedation	0.185 (0.035–0.988)			
	General anesthesia	0.453 (0.164–1.250)			
**Tumor-related characteristics**					
Margin size	<5 mm	Reference	0.008	Reference	0.138
	>5 mm	0.221 (0.07–0.679)		0.384 (0.109–1.359)	

* = intraoperative ultrasound or CT fluoroscopy, HR = hazard ratio, 95% CI = 95% confidence interval, ASA = American Society of Anesthesiologists score, NA = insufficient group comparison, RFA = radiofrequency ablation, MWA = microwave ablation, CTHA = CT hepatic arteriography. Using backward selection procedure, results of step by step.

## Data Availability

The data presented in this study are available on request from the corresponding author.

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
