# Peer review of "Efficacy of Thermal Ablation for Small-Size (0–3 cm) versus Intermediate-Size (3–5 cm) Colorectal Liver Metastases: Results from the Amsterdam Colorectal Liver Met Registry (AmCORE)"

_cancers, 2023, doi:10.3390/cancers15174346_

Round 1
Reviewer 1 Report
Broadly speaking, this work is well conceptualized and focuses on a topic of great interest and actuality in gastro-intestinal oncology.
There are just a few comments I would like to propose. To begin with, the Authors state that all procedures where at least one intermediate-size tumor was treated were included in the intermediate-size group. This implies that a patient with several (up to 4) treated small-size lesions and just one treated intermediate-size lesion be assigned to the intermediate-size group.
I speculate this classification criterion may represent a confounding factor and jeopardize the reproducibility of their results, especially when it comes to overall survival analysis, as prognosis is not only influenced by the size, but also by the number of colorectal liver metastases.
To this end, I would kindly invite the Authors to further clarify the methodology according to which they partitioned their patients into the small- and intermediate-size groups and specify the exact numbers of small- and intermediate-size tumors respectively treated in each patient assigned to the intermediate-size group.
The Authors also report that the total number of treated tumors was significantly higher in the small-size groups. I wonder whether this difference, for the aforementioned reasons, may constitute an additional flaw in the design of the comparison.
I would therefore invite the Authors to emphasize the data regarding local tumor progression-free survival (LTPFS) and local tumor control (LC) and conversely mitigate their claims regarding comparative overall survival of small-size vs. intermediate-size groups.
Minor corrections should be applied to the simple summary, where some typos have been left over. Lastly, I would not employ the term “endpoint” due to the retrospective design of this study.
None.
Author Response
REVIEWER 1
Comments and Suggestions for Authors:
Broadly speaking, this work is well conceptualized and focuses on a topic of great interest and actuality in gastro-intestinal oncology.
There are just a few comments I would like to propose. To begin with, the Authors state that all procedures where at least one intermediate-size tumor was treated were included in the intermediate-size group. This implies that a patient with several (up to 4) treated small-size lesions and just one treated intermediate-size lesion be assigned to the intermediate-size group.
I speculate this classification criterion may represent a confounding factor and jeopardize the reproducibility of their results, especially when it comes to overall survival analysis, as prognosis is not only influenced by the size, but also by the number of colorectal liver metastases.
To this end, I would kindly invite the Authors to further clarify the methodology according to which they partitioned their patients into the small- and intermediate-size groups and specify the exact numbers of small- and intermediate-size tumors respectively treated in each patient assigned to the intermediate-size group.
The Authors also report that the total number of treated tumors was significantly higher in the small-size groups. I wonder whether this difference, for the aforementioned reasons, may constitute an additional flaw in the design of the comparison.
I would therefore invite the Authors to emphasize the data regarding local tumor progression-free survival (LTPFS) and local tumor control (LC) and conversely mitigate their claims regarding comparative overall survival of small-size vs. intermediate-size groups.
Answer:
We thank the reviewer for his/her assessment and we agree that the higher number of metastases in the small-size group is a potential confounder, especially because having multiple CRLM is a known risk factor for worse survival even though partially corrected by a lower volumetric disease load for smaller-size lesions compared to intermediate size lesions. To further assess its relevance, we have conducted a univariable analysis on this potential confounder with regards to overall survival.
We have added the following to our results section (253-255): Though a higher number of CRLM was present in the small-size group, univariable analysis did not identify number of CRLM as potential confounder regarding OS (p = 0.84).
We have added the following to our discussion (385-391): The concomitant presence of small-size CRLM in many intermediate-size group patients and the significantly higher number of CRLM in the small-size group may pose confounders for survival due to the fact that prognosis is not only influenced by size, but also by the number of CRLM. However, since both size and number represent parameters to quantify volumetric disease burden, this confounder is at least partially nullified given the inverse correlation between size and volume.
We have clarified the methodology section (96-99): Patients with at least 1 tumor >3cm and ≤5 cm were included in the intermediate-size (3-5cm) group, regardless of the concomitant presence of additional small-size CRLM. Pa-tients with merely ablations for smaller-size tumors were included in the small-size (0-3cm) group.
Minor corrections should be applied to the simple summary, where some typos have been left over.
Simple summary corrected.
Lastly, I would not employ the term “endpoint” due to the retrospective design of this study.
We understand this reviewers preference to reserve the word endpoint for prospective trials. However, conformal to the consensus document on time to event endpoints (Puijk et al) we do suggest to adhere to using the term endpoints for outcome parameters such as OS, LC and LTPFS in comparative retrospective series such as the work presented. We hope this reviewer can agree to that.
Reviewer 2 Report
Peer Review Report
Manuscript ID: Cancers-2566107
Title: Efficacy of Thermal Ablation for Small-size (0-3cm) versus Intermediate-size (3-5cm) Colorectal Liver Metastases: Results from the Amsterdam Colorectal Liver Met Registry (AmCORE)
The study “Efficacy of Thermal Ablation for Small-size (0-3cm) versus Intermediate-size (3-5cm) Colorectal Liver Metastases: Results from the Amsterdam Colorectal Liver Met Registry (AmCORE)” by authors Dijkstra et al. lies within the Journal scope of Cancers. The study addressed the efficacy of thermal ablation for small size colorectal liver metastases (CRLM) (≤ 3 cm) versus intermediate size CRLM. The study summarizes results of 280 patients with 347 procedures reported between December 2000 and November 2021. The study concluded that thermal ablation is a viable option for intermediate-size unresectable CRLM tumors. There are several suggestions for authors to incorporate before recommending or reconsider the work for publication. The authors should pay attention to all aspects and given comments in next submission and ensure that all points should be adequately addressed.
1. In Simple Summary section: Please include the abbreviation next to its usage. For example: CRLM, OS, LC cannot be understood by readers until they read Abstract. Either remove or improve this section. In Line 24, the sentence structure seems incomplete: Per tumor analysis showed that……..?? (showed what??)
2. Line 44: What do you mean by COLLISION-XL trial here? There should be a brief context to explain this.
3. Lines 61-65: A recent study [https://doi.org/10.1016/j.cmpb.2020.105781] using medical imaging discussed the role of sparing healthy tissue surrounding tumors using thermal ablation. Patient-specific irregular tumor was extracted and peripheral fringe heating (tumor margins at interface of healthy tissue) were restricted to 5 mm region. Include such discussion here. Also, discuss briefly regarding the influence of blood perfusion heterogeneity of tumor during thermal ablation.
Un-resectable disease is herein defined as inability to obtain R0 margins, inability to spare sufficient future liver remnant volume and function, reduced general health status and/or major cardiopulmonary comorbidities or presumed extensive adhesions caused by previous abdominal surgery [24]
4. Lines 65-68: The authors mentions radiofrequency ablation, microwave ablation as the only thermal ablation techniques, however, there are other thermal energy delivering techniques to ablate the biologically diseased tissues such as magnetic nanoparticles induced thermal ablation [https://doi.org/10.1115/1.4046967; https://doi.org/10.1016/j.icheatmasstransfer.2021.105393], gold nanoparticles induced thermal ablation [], laser ablation [] etc. Consider including the suggested literature and then build plot that clinically the usage of microwave/radiofrequency induced ablation is performed.
5. Lines 110-113: Briefly discuss the role of blood perfusion while attaining tumor-free ablation in reference to [https://doi.org/10.1016/j.cmpb.2020.105781] using computational simulations and restrict thermal ablation to tumors. Discuss this study briefly here for tumor free ablation margins upto 10 mm region. Also, discuss in reference to this study that thermal ablation can be used as a standalone technique while minimizing damage to healthy tissues. The thermal damage ablation metric was explained in this work and may be used by researchers to quantify the margins. The thermal damage of healthy tissues with heat conduction must not be twice the volume of tumor. Discuss this here?
The intended minimum tumor free ablation margin was >1cm and the minimum realized tumor-free ablation margin to claim technical success was 5mm as calculated on confirmation software with rigid 3D image-registration (Syngo Fusion, Siemens, Erlangen, Germany).
6. Line 129: What do you mean by ceCT scan here? Please proof-read your manuscript to avoid such errors in final version. I think the authors may infer CECT (Contrast-enhanced Computed Tomography) scan here. Add such abbreviations next to such mentioning in the text.
7. Discuss the reasoning behind using Pearson chi-square test and Fisher’s exact test. There are other statistical tests also. Why do the authors only use these tests to quantify their results? Discuss your results with mean and two standard errors of mean? What is the difference between independent t-test or Mann-Whitney U test? Kaplan Meir curves with log-rank test. Cox-proportional hazards regression models? Tabulate this information?
8. Improve the resolution of figure 1. It is blurry.
9. Line 195-196: There is a huge variation in small size-group and intermediate size group? Explain?
A total of 783 tumors were included in the small size group and 73 tumors in the intermediate-size group.
10. Line 201-204: How did you compute margins in both groups?
The vast majority of ablation zones of small-size tumors showed margins 201 >5mm (94.2%), whereas for intermediate-size tumors only 58.5% reached margins >5mm 202 (p = 0.020). Median tumor size in the small-size group was 13.0mm (IQR 8.0-20.0) and in 203 the intermediate-size group 36.0mm (IQR 33.0-40.5).
11. Overall survival seems to exponentially decrease after taking thermal ablation? How do you explain these results? Provide possible reasons for such trends. Compare and contrast small size and intermediate size for both figure 2. Consider adding discussion and what inference you got from your results?.
12. Recent studies suggests change in interstitial space (porosity) of tissue during thermal ablation [https://doi.org/10.1016/j.icheatmasstransfer.2021.105393]. Interstitial space of tissue is changed during heating and it also modifies the heating requirements of tumor based on tumor vasculature. Explain and discuss in discussion section?
13. Improve Conclusion section. Provide more context and report quantitative results with qualitative inferences.
14. Discussion of results is not clear. Extensive references were cited rather than discussing the results. Improvise this section. I will re-assess this section specifically in next review.
We are looking forward to receiving your revised submission.

Please explain abbreviations next to its mention. Avoid any technical jargons in writing.
Author Response
REVIEWER 2
Manuscript ID: Cancers-2566107.
- Simple Summary section: Please include the abbreviation next to its usage. For example: CRLM, OS, LC cannot be understood by readers until they read Abstract. Either remove or improve this section. In Line 24, the sentence structure seems incomplete: Per tumor analysis showed that……..?? (showed what??)
We thank the reviewer for his/her assessment. Abbreviations and line 24 were corrected..
- Line 44: What do you mean by COLLISION-XL trial here? There should be a brief context to explain this.
Corrected and explained the ongoing COLLISION-XL trial for intermediate-size CRLM.
- Lines 61-65: A recent study [https://doi.org/10.1016/j.cmpb.2020.105781] using medical imaging discussed the role of sparing healthy tissue surrounding tumors using thermal ablation. Patient-specific irregular tumor was extracted and peripheral fringe heating (tumor margins at interface of healthy tissue) were restricted to 5 mm region. Include such discussion here. Also, discuss briefly regarding the influence of blood perfusion heterogeneity of tumor during thermal ablation.
Un-resectable disease is herein defined as inability to obtain R0 margins, inability to spare sufficient future liver remnant volume and function, reduced general health status and/or major cardiopulmonary comorbidities or presumed extensive adhesions caused by previous abdominal surgery [24]
Added the following to the discussion (366-369):
For larger-size tumors, the preferred size of the tumor-free ablation zone is not only a trade-off between efficacy and safety, but also requires taking into account tumor perfusion, tumor boundaries, intertstitial space porosity during heating and the applied heat dosage to spare healthy surrounding parenchyma.
Referring to Quantitative evaluation of effects of coupled temperature elevation, thermal damage, and enlarged porosity on nanoparticle migration in tumors during magnetic nanoparticle hyperthermia - ScienceDirect and Pre-operative Assessment of Ablation Margins for Variable Blood Perfusion Metrics in a Magnetic Resonance Imaging Based Complex Breast Tumour Anatomy: Simulation Paradigms in Thermal Therapies - ScienceDirect
- Lines 65-68: The authors mentions radiofrequency ablation, microwave ablation as the only thermal ablation techniques, however, there are other thermal energy delivering techniques to ablate the biologically diseased tissues such as magnetic nanoparticles induced thermal ablation [https://doi.org/10.1115/1.4046967; https://doi.org/10.1016/j.icheatmasstransfer.2021.105393], gold nanoparticles induced thermal ablation [], laser ablation [] etc. Consider including the suggested literature and then build plot that clinically the usage of microwave/radiofrequency induced ablation is performed.
We kindly want to point out to the reviewer that we do not mention MWA and RFA as the only thermal ablation techniques. In contrast, we describe MWA and RFA as the most utilized and researched thermal ablative energies for the treatment of colorectal liver metastases. As it would require including dozens of alternative thermal and non-thermal based ablative energies described in literature we hope this reviewer can agree not to include all ablative entities. To emphasize…
Introduction (68-71):
The most utilized and researched thermal ablative energies are radiofrequency ablation (RFA) and microwave ablation (MWA), whereas evidence for non-thermal ablative methods such as irreversible electroporation (IRE) and stereotactic ablative body radiotherapy (SABR) is piling.
- Lines 110-113: Briefly discuss the role of blood perfusion while attaining tumor-free ablation in reference to [https://doi.org/10.1016/j.cmpb.2020.105781] using computational simulations and restrict thermal ablation to tumors. Discuss this study briefly here for tumor free ablation margins up to 10 mm region. Also, discuss in reference to this study that thermal ablation can be used as a standalone technique while minimizing damage to healthy tissues. The thermal damage ablation metric was explained in this work and may be used by researchers to quantify the margins. The thermal damage of healthy tissues with heat conduction must not be twice the volume of tumor. Discuss this here?
Added/changed in the M&M section (118-123):
Conformal to the CIRSE standards of practice on thermal ablation of liver tumors, the intended minimum tumor free ablation margin was >1cm and the minimum realized tumor-free ablation margin to claim technical success was 5mm [37, 38]. Ablation zone margins were calculated on confirmation software with rigid 3D image-registration (Syngo Fusion, Siemens, Erlangen, Germany) directly after the ablation.
Added/changed in the discussion (366-369):
For larger-size tumors, the preferred size of the tumor-free ablation zone is not only a trade-off between efficacy and safety, but also requires taking into account tumor perfusion, tumor boundaries, interstitial space porosity during heating and the applied heat dosage to spare healthy surrounding parenchyma.
- Line 129: What do you mean by ceCT scan here? Please proof-read your manuscript to avoid such errors in final version. I think the authors may infer CECT (Contrast-enhanced Computed Tomography) scan here. Add such abbreviations next to such mentioning in the text.
Yes ceCT stands for contrast enhanced CT similarly to CECT as the reviewer points out. Abbrevation explained accordingly in the M&M section.
- Discuss the reasoning behind using Pearson chi-square test and Fisher’s exact test. There are other statistical tests also. Why do the authors only use these tests to quantify their results? Discuss your results with mean and two standard errors of mean? What is the difference between independent t-test or Mann-Whitney U test? Kaplan Meir curves with log-rank test. Cox-proportional hazards regression models? Tabulate this information?
The statistical analysis of this study was both designed and supervised by our co-authored biostatistian. To compare dichotomous or nominal variables between two groups, Pearson chi-square and Fisher’s exact test are the appropriate tests to use. When comparing two groups using mean, you need a continues variable (for example age and BMI), there we used T-test. Independent samples t-test was used when variables were normally distributed and Mann–Whitney U Test was used when not-normally distributed for continuous variables. The Kaplan–Meier method using the log-rank test was used to analyze time-to-event variables. To compare and account for potential variables between the two groups Cox proportional hazards regression models with multivariable analysis was used.
- Improve the resolution of figure 1. It is blurry.
Figure 1 was replaced, and all tables are provided in word version.
- Line 195-196: There is a huge variation in small size-group and intermediate size group? Explain? A total of 783 tumors were included in the small size group and 73 tumors in the intermediate-size group.
This can be explained by the circumstance that thermal ablation for unresectable, small-size tumors is already extensively endorsed within international guidelines. In contrast to unresectable intermediate-size tumors, where tumors are often first treated with chemotherapy to downsize and eventually ablate them as small-size CRLM. One of the goals of this study was to assess efficacy of thermal ablation in unresectable intermediate size tumors, which is currently outside of most (inter)national guideline indications.
Added to the introduction (72-79):
Increased tumor sizes (>3cm) are associated with exponentially reduced technical ef-ficacy and shorter local tumor progression (LTP) free survival [25-31]. A recent systematic review and meta-analysis by van Nieuwenhuizen et al. compared safety and efficacy of thermal ablation, IRE and SABR for intermediate-size CRLM (3-5cm) [32]. Per-patient local control ranged 22-89% (in 8 series) following thermal ablation and results improved over time. Nonetheless, thermal ablation for unresectable intermediate-size tumors, is currently still outside most of the international guidelines.
- Line 201-204: How did you compute margins in both groups?
The vast majority of ablation zones of small-size tumors showed margins 201 >5mm (94.2%), whereas for intermediate-size tumors only 58.5% reached margins >5mm 202 (p = 0.020). Median tumor size in the small-size group was 13.0mm (IQR 8.0-20.0) and in 203 the intermediate-size group 36.0mm (IQR 33.0-40.5).
Margins are calculated by the interventional radiologist, directly after the ablation with confirmation software using rigid 3D image-registration of the immediately pre-and post-ablation ceCT scans (Syngo Fusion, Siemens, Erlangen, Germany) with minimum ablation margins eventually stated per ablated tumor in the EPD using smartphrases and noted in the prospective AmCORE registry.
We have added the following to the M&M section (118-123):
Conformal to the CIRSE standards of practice on thermal ablation of liver tumors, the intended minimum tumor free ablation margin was >1cm and the minimum realized tumor-free ablation margin to claim technical success was 5mm [37, 38]. Ablation zone margins were calculated with confirmation software with rigid 3D image-registration (Syngo Fusion, Siemens, Erlangen, Germany) directly after the ablation.
- Overall survival seems to exponentially decrease after taking thermal ablation? How do you explain these results? Provide possible reasons for such trends. Compare and contrast small size and intermediate size for both figure 2. Consider adding discussion and what inference you got from your results?.
Survival times were defined according to the DATECAN consensus document as follows: the time evolved between the starting time (the date of the (first) ablative procedure) and the time of the last follow-up or the time of a certain event (in OS obviously death). Death without having detected a certain event (such as LTP or loss of LC) was reported as a competing risk and censored when using Kaplan-Meier estimates. The sigmoidal survival curves with an eventual plateau in our opinion compares well to survival curves in other series on local treatments for metastatic colorectal cancer. The 1-, 3- and 5-year OS rates as well as the median OS (5 years) is good compared to most other series in literature on thermal ablation as is the 15-20% surviving >10 years following the ablation.
We agree that the higher number of metastases in the small-size group is a potential confounder, especially because having multiple CRLM is a known risk factor for worse survival even though partially corrected by a lower volumetric disease load for smaller-size lesions compared to intermediate size lesions. To further assess its relevance, we have conducted a univariable analysis on this potential confounder with regards to overall survival.
We have added the following to our results section (253-255): Though a higher number of CRLM was present in the small-size group, univariable analysis did not identify number of CRLM as potential confounder regarding OS (p = 0.84).
We have added the following to our discussion (385-391): The concomitant presence of small-size CRLM in many intermediate-size group patients and the significantly higher number of CRLM in the small-size group may pose confounders for survival due to the fact that prognosis is not only influenced by size, but also by the number of CRLM. However, since both size and number represent parameters to quantify volumetric disease burden, this confounder is at least partially nullified given the inverse correlation between size and volume.
- Recent studies suggests change in interstitial space (porosity) of tissue during thermal ablation [https://doi.org/10.1016/j.icheatmasstransfer.2021.105393]. Interstitial space of tissue is changed during heating and it also modifies the heating requirements of tumor based on tumor vasculature. Explain and discuss in discussion section?
Added to the discussion (366-369):
For larger-size tumors, the preferred size of the tumor-free ablation zone is not only a trade-off between efficacy and safety, but also requires taking into account tumor perfusion, tumor boundaries, interstitial space porosity during heating and the applied heat dosage to spare healthy surrounding parenchyma.
- Improve Conclusion section. Provide more context and report quantitative results with qualitative inferences.
Conclusion section changed accordingly.
- Discussion of results is not clear. Extensive references were cited rather than discussing the results. Improvise this section. I will re-assess this section specifically in next review.
Extensively revised.
The authors want to thank the reviewer for taking the time and attention he has taken to evaluate our discussion. We hope to have sufficiently answered your questions
Round 2
Reviewer 2 Report
There is a typo in Line 403 of revised version of Manuscript under Conclusions section 5.0. It should be Results. Please proof-read your work one time more.
esults from randomized controlled trials such as the COLLISION-XL trial
The authors must proof-read their work as I have identified some typos in the writing.